# Interactions of Different *Streptomyces* Species and *Myxococcus xanthus* Affect *Myxococcus* Development and Induce the Production of DK-Xanthenes

**DOI:** 10.3390/ijms242115659

**Published:** 2023-10-27

**Authors:** Ramón I. Santamaría, Ana Martínez-Carrasco, José R. Tormo, Jesús Martín, Olga Genilloud, Fernando Reyes, Margarita Díaz

**Affiliations:** 1Instituto de Biología Funcional y Genómica (IBFG), Departamento de Microbiología y Genética, Consejo Superior de Investigaciones Científicas (CSIC), Universidad de Salamanca, C/Zacarías González, nº 2, 37007 Salamanca, Spain; amcgar@idaea.csic.es; 2Fundación MEDINA, Centro de Excelencia en Investigación de Medicamentos Innovadores en Andalucía, Avda. del Conocimiento 34, 18016 Granada, Spain; ruben.tormo@medinaandalucia.es (J.R.T.); jesus.martin@medinaandalucia.es (J.M.); olga.genilloud@medinaandalucia.es (O.G.); fernando.reyes@medinaandalucia.es (F.R.)

**Keywords:** *Streptomyces*, *Myxococcus*, co-cultures, siderophores, nocardamine, demethylenenocardamine, morphological development

## Abstract

The co-culturing of microorganisms is a well-known strategy to study microbial interactions in the laboratory. This approach facilitates the identification of new signals and molecules produced by one species that affects other species’ behavior. In this work, we have studied the effects of the interaction of nine *Streptomyces* species (*S. albidoflavus*, *S. ambofaciens*, *S. argillaceus*, *S. griseus*, *S. lividans*, *S. olivaceus*, *S. parvulus*, *S. peucetius*, and *S. rochei*) with the predator bacteria *Myxococcus xanthus*, five of which (*S. albidoflavus*, *S. griseus*, *S. lividans*, *S. olivaceus*, and *S. argillaceus*) induce mound formation of *M. xanthus* on complex media (Casitone Yeast extract (CYE) and Casitone tris (CTT); media on which *M. xanthus* does not form these aggregates under normal culture conditions. An in-depth study on *S. griseus*–*M. xanthus* interactions (the *Streptomyces* strain producing the strongest effect) has allowed the identification of two siderophores produced by *S. griseus*, demethylenenocardamine and nocardamine, responsible for this grouping effect over *M. xanthus*. Experiments using pure commercial nocardamine and different concentrations of FeSO_4_ show that iron depletion is responsible for the behavior of *M. xanthus*. Additionally, it was found that molecules, smaller than 3 kDa, produced by *S. peucetius* can induce the production of DK-xanthenes by *M. xanthus*.

## 1. Introduction

The use of axenic microbial cultures is the traditional method to study microorganisms in the laboratory, and the expression of the genomes of some of them has been established under different culture conditions. These experiments have led to the conclusion that in *Streptomyces* in particular, and other bacteria and fungi in general, a large number of genes and clusters encoding complex compounds remain silent or are poorly expressed under laboratory conditions [1].

Co-culture experiments attempt to mimic, in some way, unique ecological conditions to study whether the interaction between different organisms triggers the expression of silent genes, activating the production of interesting complex compounds [2]. Thus, over the last decades, several publications have focused on this topic, demonstrating that this is a viable strategy to increase the production of different compounds and/or the diversity of the compounds produced by partners compared to their axenic cultures [2]. Notably, different co-cultures have been implemented to generate products that cannot be obtained through the fermentation of one single species [3]. Moreover, interest in co-culturing has triggered the development of new technological devices that can help to identify the signals used by microorganisms to communicate [4].

Bacteria of the *Streptomyces* genus are soil-borne bacteria and the genome of more than 600 strains has been sequenced [5]. The in silico analysis of these sequences revealed the occurrence of 20 to 50 different biosynthetic pathways that might direct the biosynthesis of a wide diversity of specialized metabolites bearing different biological activities such as antimicrobial, antitumor, pheromones, and siderophores, among others [6]. However, many of the pathways present are not expressed under laboratory conditions and the conditions triggering their expression remain to be discovered [7,8]. Activation of some of these pathways has, however, been described as a result of the interaction mediated by a compound produced by another microorganism such as bacteria, fungi, or higher superior organisms such as insects and plants [9,10]. Several examples of *Streptomyces* interactions have been reported so far and, among them, we have selected a few to illustrate their potential. For instance, Onaka et al., in 2011, reported that the interaction of *Streptomyces lividans* with mycolic acid-containing bacteria, such as *Tsukamurella pulmonis* and other members of the the *Corynebacteriaceae* family, induces the production of red pigments, putatively prodiginine. Furthermore, the production of a new antibiotic, alchivemycin A, was triggered when *Streptomyces lividans* and *T. pulmonis* were co-cultured [11,12]. Actinorhodin production by *Streptomyces coelicolor* was stimulated when cultured close to *M. xanthus*, and this response was due to increased production by *M. xanthus* of a siderophore, myxochelin, which indicates iron competition between the two organisms [13,14]. This iron restriction increases the expression of 21 secondary metabolite biosynthetic gene clusters (smBGCs) in other *Streptomyces* species [13,15]. The production of at least 12 variants of desferrioxamine, a siderophore used by the bacteria to capture iron, has been described in work studying the interactions between different *Streptomyces* species [16,17].

Co-culture studies on *Streptomyces* in the presence of marine bacteria have led to the observation that the time of initiation of the interaction is important for the response obtained [18]. The production of a novel cyclic hexapeptide antibiotic, incorporating three piperazine acids called dentigerumycin E, has been described as a result of interaction between a marine *Streptomyces* sp. and a *Bacillus* sp. [19]. The production of gordonic acid, a novel polyketide glycoside, from a co-culture of *Streptomyces tendae* KMC006 and *Gordonia* sp. KMC005 has also been reported [20]. Maglangit et al. described that the interaction of a *Streptomyces* sp. MA37 strain with *Pseudomonas* sp. induces the expression of a cryptic pathway encoding a bioactive indole alkaloid metabolite [21].

Several studies on the interactions of different *Streptomyces* with various fungi have shown that co-cultures with species of the genus *Aspergillus* are the most prolific. New molecules, such as N-formyl alkaloids, diketopiperazine alkaloids, fumicyclines, and others, are produced by *A. fumigatus* when in close contact with *S. peucetius*, *S. bullii*, or *S. rapamycinicus*, respectively [15,22,23]. The production of orsellinic acid and lecanoric acid is detected when *A. nidulans* and *S. ramamycinicus* are grown together [24,25]. In co-cultures of the marine-derived fungal isolate *A. fumigatus* MR2012 and two hyper-arid desert bacterial isolates, *S. leeuwenhoekii* strain C34 and strain C58, a concomitant induction of newly detected bacterial and fungal metabolites was obtained in both organisms [15]. Recently, Nicault et al. analyzed 72 interactions among 8 different *Streptomyces* strains and 9 different fungi and described that two of these interactions, studied in more detail, have a dramatic impact on the metabolic expression of each partner [26].

In addition to the induction of secondary metabolites, other phenomena occur when more than one species is present in a culture. For example, changes in the developmental program of *Streptomyces* was observed in relation to the production of a lipopeptide, surfactin, by *Bacillus subtilis* [27]. Another interesting study describes that the interaction of *Streptomyces venezuelae* and *Saccharomyces cerevisiae* leads to the generation of *Streptomyces* explorer cells due to the change in iron availability [28]. A comprehensive review on *Streptomyces* interactions with other microorganisms was previously published by Kim et al., 2021 [29].

In this work, we examine the interaction of different *Streptomyces* species with the Gram-negative predatory bacterium *M. xanthus*. *M. xanthus* displays a complex life cycle in which normal bacillary cells coordinate their motility and, under starvation conditions, aggregate in large fruiting bodies where the cells differentiate into spherical spores [30]. This multicellular development is also driven by interactions between *M. xanthus* cells and their cognate prey [31].

*M. xanthus* is a predatory bacterium that has 8.5% of its genome devoted to the production of secondary metabolites [32]. Hence, its genome has 24 biosynthetic gene clusters for secondary metabolites, some of which encode antibacterial and antifungal compounds [33]. Efficient predation by *M. xanthus* requires two types of motility: social motility (S) and adventurous motility (A) which allows the predator to stay within the area and synchronize its movement by generating oscillating waves, known as rippling, that increase the expansion of the predator [14,34]. *M. xanthus* can recognize cells of other bacterial microorganisms and can discriminate live from dead cells. In this regard, Livingstone et al. studied the transcriptome changes associated with its predation on live or dead *Escherichia coli*. They demonstrated that exposure to dead prey significantly alters the expression of 1319 *M. xanthus* predator genes, whereas the transcriptional response to live prey was minimal, with only 12 genes being significantly upregulated [35]. In another interesting paper, the effect of metals on *M. xanthus* predation was studied, showing that copper plays an important role during *M. xanthus* predation on *Sinorhizobium meliloti*. In this context, melanin is used by the prey to defend itself against the predator [36].

In previous work, we showed that the interaction between *S. coelicolor* and *M. xanthus* increases actinorhodin production by *S. coelicolor* [14]. In this study, we have broadened the co-culture analysis of *Streptomyces* sp./*M. xanthus* to include nine additional *Streptomyces* species obtained from different collections. This work aimed to test whether these interactions could activate silent pathways, increase the production of known metabolites in any of the bacteria used in the interactions, and determine whether these nine species could induce changes in the development of *M. xanthus*.

## 2. Results

### 2.1. Interactions between Various Species of Streptomyces and M. xanthus Exert Different Responses Depending on the Streptomyces Species Assayed

To study the effect of the co-culture interactions of *Streptomyces* sp./*M. xanthus*, drops (containing 5 × 10^6^
*Streptomyces* spores, or mycelia fragments for *S. peucetius*) of different *Streptomyces* species (*S. albidoflavus*, *S. ambofaciens*, *S. argillaceus*, *S. griseus*, *S. lividans*, *S. olivaceus*, *S. parvulus*, *S. peucetius*, and *S. rochei*) were deposited on the surface of solid Casitone-yeast extract medium (CYE) close to *M. xanthus* DK1622 drops (10^6^ cells). Two individual spots of *M. xanthus* DK1622 and two individual spots of each *Streptomyces* strain were used as control. The plates were incubated at 28 °C for 7–12 days.

As observed in Figure 1, mounds of *M. xanthus* aggregates were induced by five of the *Streptomyces* species studied, *S. albidoflavus*, *S. griseus*, *S. lividans*, *S. olivaceus*, and *S. argillaceus.* At the same time, four of these species, S. *albidoflavus*, *S. griseus*, *S. lividans*, and *S. olivaceus*, were extremely sensitive to *M. xanthus* attack that grew on and/or around the *Streptomyces* spot. However, in other interactions, such as with *S. argillaceus* and *S. parvulus*, the lytic effect produced by *M. xanthus* over these species was somewhat limited. Although *M. xanthus* growth was abundant, it was unable to surround the *Streptomyces* spots. In other interactions, such as with *S. peucetius* and *S. rochei*, the lytic effect on these *Streptomyces* strains was extremely limited, i.e., *M. xanthus* moved away from the spot containing *S. peucetius*. Finally, poor *M. xanthus* growth was observed upon its interaction with *S. ambofaciens*. This indicated that this *Streptomyces* strain produces a molecule(s) able to significantly limit the growth of *M. xanthus* (Figure 1). Control spots of the different species of *Streptomyces* are shown in Appendix A. These experiments were also performed on solid Casitone Tris (CTT) medium, and the same effect was observed, as seen in Appendix A.

### 2.2. M. xanthus Aggregates Are Formed When It Is Co-Cultured with Different Streptomyces Species

In our experiments, aggregates of *M. xanthus* were observed in interactions with *S. albidoflavus*, *S. griseus*, *S. lividans, S. olivaceus*, and *S. argillaceus* as dense spots on the complex medium CYE (Figure 1). Therefore, one of the goals of this work was to determine the signal/s, possibly secreted in the culture medium, responsible for the observed group formation induced during the *M. xanthus/Streptomyces* interactions.

Of the nine species tested, the grouping effect exerted by *S. griseus* was the most dramatic. Consequently, we focused on this interaction to identify a possible inducer molecule. The aim of the first experiment was to determine whether the *S. griseus* inducer was also produced and secreted in liquid media. Therefore, filtered supernatants of this strain, grown in different liquid media (CYE, R2YE, and R5A-sucrose) for 7 days, were tested by adding each one separately to a well adjacent to an *M. xanthus* spot. It was observed that all the supernatants were able to induce the formation of mounds by *M. xanthus*. However, the supernatant that originated from the R5A-sucrose culture was the most efficient at inducing the formation of *Myxococcus* aggregates on solid CYE. Thus, this supernatant was selected for conducting further studies.

Since some reports mentioned that glycerol or inducers of beta-lactamase, such as ampicillin, could induce the mound formation of *M. xanthus* in complex media [37,38,39], we also studied whether these compounds induced aggregation under our culture conditions. Two different concentrations of each compound were used, 0.5 and 1 M of glycerol and 2 and 5 mM of ampicillin; H_2_O was used as the negative control. None of these conditions was able to induce the aggregation of *M. xanthus* as efficiently as the supernatant of *S. griseus* (Figure 2A). Scanning electronic microscopy showed that the mounds generated in the presence of the *S. griseus* supernatant presented some myxospores that were absent in the negative control (Figure 2B).

To identify the putative molecule(s) present in the *S. griseus* supernatant able to induce this social behavior, we used Amicon centrifugal filters (Millipore) of different pore sizes (30, 10, and 3 kDa) to fractionate the supernatant of *S. griseus* when grown in R5A-sucrose medium. The different fractions obtained (eluted [E] and retained [R] with each pore size used) were assayed against *M. xanthus* DK1622. These experiments indicated that the putative inducer molecule(s) was smaller than 3 kDa because it was not retained even when using a filter with a 3 kDa pore size (Figure 3A). This sample was fractionated by reversed-phase HPLC to generate 80 subfractions, as indicated in the Experimental Procedures (Appendix A), that were evaluated for their activity against *M. xanthus* DK1622. Out of the 80 subfractions, fractions F30 and F31 induced the grouping effect (Figure 3B).

The analysis of the components of these two subfractions by High-Resolution Mass Spectrometry and fingerprinting comparisons, using databases available through the Fundación MEDINA of more than 2150 Natural product standards [40], identified that they contained demethylenenocardamine and nocardamine, respectively. Both molecules are siderophores produced by *Streptomyces* that capture iron with high affinity. To confirm the effect of these compounds on *M. xanthus* grouping, we performed assays with different concentrations of commercial nocardamine (deferrioxamine E). The grouping behavior of this compound was clearly observed at concentrations higher than 10 μM (Figure 4).

To corroborate that *M. xanthus* grouping was due to the sequestration of iron from the media by the siderophore nocardamine, we added different amounts of FeSO_4_ (from 2.5 to 25 μM) to CYE medium, with or without 50 μM of nocardamine, deposited in a well.

The grouping effect was observed in plates containing up to 12.5 μM of FeSO_4_, but not at higher concentrations (Figure 5). Therefore, the grouping effect of *M. xanthus* observed in the co-culture of *S. griseus* and *M. xanthus* is due to the sequestration of iron present in the media by the nocardamine secreted into the culture medium by *S. griseus*.

Two other commercial siderophores, deferoxamine mesylate salt (DFOM) and 2,2′-bipyridyl, were also evaluated. The grouping behavior induced by deferoxamine mesylate salt was observed with concentrations as high as 20 μM, while for 2,2′-bipyridyl (BP), the grouping behavior was observed when using concentrations higher than 100 μM (Appendix A).

### 2.3. Identification of Molecules Produced in the Interactions between Different Species of Streptomyces and M. xanthus

In addition to the induction of aggregates formation by *M. xanthus*, the induced production of natural products through the interaction of *M. xanthus* with the different *Streptomyces* species studied was also analyzed. UPLC analyses of the organic extracts obtained from pieces of solid CYE agar, containing the areas where the different organisms interacted, and the corresponding axenic culture controls allowed us to corroborate that only the *S. peucetius*/*M. xanthus* co-culture induced the production of several molecules (Appendix A). None of the other co-cultures tested showed a clear induction of novel molecules different from those produced in axenic cultures of the different *Streptomyces* species assayed or *M. xanthus* cultures (Appendix A).

To determine whether the effect observed between *S. peucetius* and *M. xanthus* on solid medium co-cultures also occurred in liquid media, these microorganisms were grown together in liquid CYE for three days at 28 °C under agitation. Axenic control cultures of *S. peucetius* and *M. xanthus* were also grown at the same time. Analysis of the organic extracts (ethyl acetate acidified) indicated that several compounds had been induced (Figure 6A). The analysis of the molecules produced in this interaction revealed that peaks P1, P2, and P3, obtained from minute 3.8 to 5.8, corresponded to different DK-xanthenes, with DK-xanthene 574 being the most abundant compound [41,42].

Since DK-xanthenes are known to have antifungal activity [41,42], the antifungal activity of the controls (*M. xanthus* and *S. peucetius*) and the *M. xanthus/S. peucetius* co-culture was checked through a bioassay using *S. cerevisiae* W303 as the sensitive strain. A clear antifungal effect was observed in the coculture of *M. xanthus/S. peucetius* but was not detected in the supernatants of the axenic cultures of *M. xanthus* and *S. peucetius* grown as controls (Figure 6B).

To explore this further, the next step was to study whether direct contact between *S. peucetius* and *M. xanthus* was necessary for induction or whether a filtrated supernatant of *S. peucetius* was able to induce the production of these molecules in *M. xanthus* cultures. Consequently, axenic cultures of *S. peucetius* were generated in CYE, R2YE, and R5A-sucrose liquid media for 7 days at 28 °C. Different amounts of filtered supernatants were added to *M. xanthus* axenic cultures in liquid CYE and maintained under agitation for 3–5 days at 28 °C. Axenic cultures of *M. xanthus* were performed in parallel as a reference. Subsequent HPLC-MS analyses of the organic extracts obtained from the different conditions showed that the *S. peucetius* supernatant originating from R5A-sucrose medium was the most active effector. Additionally, it was determined that the addition of 1.5% of filtered *S. peucetius* supernatant to *M. xanthus* CYE cultures was sufficient to induce the production of the three peaks detected in the *M. xanthus/S. peucetius* co-culture (Figure 7A).

To identify whether the antifungal activity detected in the cultures of *M. xanthus*, grown in the presence of *S. peucetius* supernatant, was due to the overproduction of DK-xanthenes, the effect of the *S. peucetius* supernatant on the wild type strain DK1050 (WT) and its derivative strain DK1050 PMΔRF_N, DK-xanthene deficient (DK^−^), was studied [43]. Cultures were grown in liquid CYE in the presence of 1.5% filtered *S. peucetius* supernatant and, after 3 days, HPLC separation of the compounds produced by both strains showed that these compounds were not produced by the DK-xanthene deficient strain induced by *S. peucetius* supernatant (Figure 7B). Antifungal activity of these extracts was detected in the wild-type strain MX1050 grown in the presence of *S. peucetius* supernatant but not in the extracts of the *M. xanthus* DK-xanthene-deficient strain grown under the same conditions. This result suggests that the antifungal activity detected in the wild type strain could be caused by the DK-xanthenes produced (Figure 7C).

### 2.4. Purification of the Inducer Molecule Present in S. peucetius Supernatant

To identify the molecule or molecules produced by *S. peucetius* responsible for the inducing effect, we used centrifugal filters, Amicon (Millipore), of different pore sizes (30, 10, and 3 kDa) to fractionate the supernatant of *S. peucetius* grown in R5A-sucrose medium. The different fractions obtained were added to liquid CYE cultures inoculated with *M. xanthus* DK1622. The supernatants of 3–5 day cultures were used in a bioassay against *S. cerevisiae* W303. The activity was detected upon using the 3 kDa pass-through fraction (E3), which indicated that the inducers were smaller than 3 kDa (Figure 8). Fraction (E3) was loaded in an HP-20 adsorption resin and the activity was only present in the non-retained aqueous SPE flow-through. Then, a size exclusion fractionation in Sephadex LH-20 with 100% water was performed using this aqueous phase where activity fractions (F4–F6) were further fractionated by preparative HPLC in a T3 Atlantis OBD column. In total, 84 fractions were collected (F1–F84) (Appendix A) and assessed for their ability to induce the production of compounds active against *S. cerevisiae*. As a result, fractions F13 to F15 were found to induce compound production. The analysis of the components of these three fractions by High-Resolution Mass Spectrometry and fingerprinting comparisons using the databases available through the Fundación MEDINA did not identify any of the compounds present in the sample. Also, using nuclear magnetic resonance spectroscopy, it could only be detected the presence of the MOPS buffering agent (3-(N-morpholino) propanesulfonic acid) in the three fractions [40]. However, experiments involving different concentrations of MOPS in *M. xanthus* cultures did not have any effect on the induction of DK-xanthenes. So, further experiments will be need to dilucidate the nature of the inducer compounds.

## 3. Discussion

In our study, five of the nine *Streptomyces* strains tested induced the grouping of *M. xanthus.* Molecular analysis of this effect, using *S. griseus* or its liquid supernatant, showed that the *S. griseus/M. xanthus* interaction induced the production of two siderophore molecules by *S. griseus*. Both iron chelators, demethylenenocardamine and nocardamine, could lead to the depletion of free iron for *M. xanthus* growth and thus could induce the formation of groups and the generation of some myxospores in these groups in response to the stress generated in complex culture media. While demethylenenocardamine has been described as being produced by only a few *Streptomyces* species, nocardamine is produced by a large number of *Streptomyces* strains and other related microorganisms [17,44,45,46]. In fact, the positive effect of nocardamine on the growth and development of *S. tanashiensis* has been previously described, and the iron-chelating ability of nocardamine produced by *Streptomyces* sp. H11809 has been shown to starve *Plasmodium falciparum* 3D7 (*Pf* 3D7) malaria parasites of their iron source, inhibiting their growth [17,47]. 

All of the results obtained for the *S. griseus/M. xanthus* interaction indicate that competition for iron, via siderophore piracy, is a normal strategy in nature, where iron compounds can alter patterns of gene expression and morphological differentiation during interactions between microorganisms [3,48].

In addition, in the experiments performed in this work, we did not detect the production of new molecules originating from the cryptic pathways present in the genomes of the nine *Streptomyces* species tested or in the genome of *M. xanthus*. Furthermore, this *S. peucetius/M. xanthus* co-culture resulted in a dramatic increase in the production of DK-xanthenes by *M. xanthus.* The same induction was observed when *S. peucetius* supernatant, obtained from liquid R5A-sucrose cultures, was added to liquid CYE or CTT cultures of *M. xanthus*. These results indicate that direct contact between the two bacteria is not necessary to achieve induction as in other interactions between different *Streptomyces* and *Aspergillus* species [25]. This inducing ability of *S. peucetius* was previously described in the production of two new compounds, fumiformamide and N,N′-((1Z,3Z)-1,4-bis(4-methoxyphenyl)buta-1,3-diene-2,3-diyl) diformamide, by *A. fumigatus* in co-cultures [23].

The production of 13 yellow pigment derivatives of DK-xanthenes, encoded by a 47 kb cluster, by *M. xanthus* DK1050, was described as being crucial for *M. xanthus* [49]. The production of four new DK-xanthenes in *M. xanthus* has been described in a comparative study of DK-xanthenes from *Myxococcus stipitatus* DSM145675 and *M. xanthus* DK1622, and the antifungal activity of DK-xanthenes isolated from both species has also been shown [42]. Interestingly, upregulation of the expression of 10 DK-xanthene genes has recently been described in *M. xanthus* predation on *Sinorhizobium meliloti* [50].

During predation, *M. xanthus* uses outer membrane vesicles (OMVs) to release lethal effectors in the proximity of prey. OMVs contain enzymes with hydrolytic activities and antibiotics such as myxovirescin A and the antifungal myxalamide [51]. But in some cases, the prey induces different defense mechanisms such as a mechanical barrier or the production of antibiotics. Thus, *B. subtilis* generates megastructures with spores that resist the attack of *M. xanthus* [52], and *S. coelicolor* overproduces the antibiotic actinorhodin which is not active against Gram-negative bacteria but putatively acts as a repellent of *M. xanthus* that moves away from the contact area of the *S. coelicolor* colony [14]. Transcriptomic changes in *M. xanthus* genes during co-culture or predation against different prey have also been analyzed [13,35,50,53]. Thus, a comparison of the “predatosome” against *S. meliloti* and against *S. coelicolor* identified 76 common genes that were upregulated and 11 genes that were downregulated with common features of modification in lipid metabolism, iron uptake, and motility [50]. Competition for iron uptake was described as the main effector in the induction of actinorhodin production by *S. coelicolor* when co-cultured with *M. xanthus.* Under these co-culture conditions, *M. xanthus* overproduces the siderophore myxochelin, which allows this bacterium to dominate iron uptake, causing the *Streptomyces* strain to have iron-restricted conditions. In fact, iron-restricted conditions increase the expression of 21 secondary metabolite biosynthetic gene clusters in other *Streptomyces* species [13]. In our study, the siderophores produced by *S. griseus* control *M. xanthus* development.

In this work, we have also observed that *S. peucetius* induces the production of DK-xanthenes, but we were not able to identify the molecule or molecules smaller than 3 kDa responsible for this effect. Different purification and chemical identification strategies were employed, but none of them led to the identification of the inducing molecule.

## 4. Materials and Methods

### 4.1. Bacterial Strains and Media

The *Streptomyces* strains used as prey in this work were: *S. albidoflavus* J1074, *S. ambofaciens* ATCC 23877, *S. argillaceus* ATCC 12596, *S. griseus* IMRU3570, *S. lividans* 1326, *S. olivaceus* Tue22, *S. parvulus* JI2283, *S. peucetius* ATCC 27952, and *S. rochei* CECT 3329. The wild-type (wt) *M. xanthus* DK1622 [54] was used as the predator. *M. xanthus* DK1050 [55] and its DK-xanthene minus strain DK1050 PMΔRF_N [43] were used to compare antifungal activity against *Saccharomyces cerevisiae*. Solid (1.5% Bacto-agar) and liquid CYE (1% Bacto-casitone, 0.5% Yeast Extract, 0.1% MgSO_4_. 7H_2_O, pH 7.6) and CTT [56] media were used to grow *M. xanthus*. R2YE and R5A-without sucrose (R5A-sucrose) were usually used for the *Streptomyces* cultures [57,58]. YEPD was used to grow *S. cerevisiae* W303 2N and for testing the antifungal activity of the compounds produced by *M. xanthus* [59]. 

### 4.2. Predation Experiments on Solid Media

The predation assays were carried out as previously indicated, with some minor modifications [14]. Briefly, 10 mL drops containing 10^6^
*M. xanthus* cells from fresh cultures were deposited on the surface of CTT or CYE agar plates and air dried. Then, 10 mL drops containing 5 × 10^6^
*Streptomyces* spores were placed close to one of the spots containing the *Myxococcus* cells (no more than 3 mm apart). In the case of *S. peucetius*, mycelium fragments were used, as, in our hands, this strain sporulates very poorly under laboratory conditions. Two spots of *M. xanthus* or two spots of the corresponding *Streptomyces* strain were deposited in the same way and used as controls. All plates were incubated at 28 °C and images were captured every two days with a digital camera under a Zeiss Stemi SV11 dissecting microscope for 2–12 days. Each experiment was repeated at least four times.

### 4.3. Study of the Effect of S. griseus on M. xanthus

The grouping effect of *S. griseus* on *M. luteus* was studied on solid CYE and CTT media by growing both organisms as indicated in the predation experiments. The effect of *S. griseus* supernatants on axenic cultures of *M. xanthus* was assessed by culturing *S. griseus* in three different liquid media: CTT, R2YE, or R5A-sucrose for 7 days. The cultures were centrifuged and the supernatants were passed through a 0.22 mm filter. Then, 200 µL of the *S. griseus* supernatants were added to a well close to the *M. xanthus* spot on solid CTT or CYE media three times with six-hour intervals between each application. The plates were incubated at 28 °C and photographs were taken from day 2 up to day 7. The formation of mounds by *M. xanthus* on CYE was also analyzed in the presence of glycerol (0.5 and 1 M) and the presence of ampicillin (2 and 5 mM).

### 4.4. Liquid Co-Cultures of S. peucetius and M. xanthus

*S. peucetius* mycelium was inoculated into 100 mL three-baffled flasks containing 10 mL of CYE and incubated under agitation (200 rpm) at 28 °C for 48 h. *M. xanthus* DK1622 was grown in 10 mL of liquid CYE at 28 °C for 24 h. Three new 10 mL cultures were grown in CYE using the previous cultures as the inoculum. Two of them were the axenic cultures of *M. xanthus* DK1622 and *S. peucetius*, used as controls, and the other was the co-culture *M. xanthus* DK1622/*S. peucetius*. These cultures were incubated at 28 °C for 3–5 days. 

The effect of *S. peucetius* supernatants on axenic cultures of *M. xanthus* was studied by culturing *S. peucetius* in three different liquid media: CYE, R2YE, or R5A-sucrose for 7 days. The cultures were centrifuged and the supernatants passed separately through a 0.22 mm filter. Different amounts of the *S. peucetius* supernatants were added to the *M. xanthus* liquid CYE cultures and maintained under shaking at 28 °C for 3–5 days. 

### 4.5. Fractionation of the Streptomyces Supernatant

Amicon centrifugal filters (Millipore, Burlington, MA, USA) of different pore sizes (30, 10, and 3 kDa) were used to initially fractionate the supernatants of *S. griseus* and *S. peucetius* grown in liquid R5A-sucrose. The different fractions were assayed against *M. xanthus* strains.

All fractions, obtained throughout the fractionation of the *S. griseus* supernatant, were assayed on solid CYE medium and the plates were monitored for the presence of *M. xanthus* mounds.

The eluted 3 kDa fraction of *S. griseus* (E3, 195 mL) was then retained in HP-20 adsorption resin (10 g) for subsequent solid phase extraction (SPE) and washed extensively with water. The retained natural products were eluted with acetone and dried under nitrogen-heated steam, producing 161 mg of a dried organic extract. Half of this material was dissolved in the minimum amount of dimethyl sulfoxide (525 µL) and fractionated by preparative reversed-phase HPLC on a Zorbax SB-C8 column from Agilent (Agilent Technologies, Santa Clara, CA, USA) (21.2 × 250 mm, 7 μm; 20 mL/min; UV detection at 210 and 280 nm) using a linear gradient of 5 to 66% acetonitrile in water for 35 min at 10 mL/min, followed by a washing step with 100% acetonitrile for 10 min. The chromatographic fractions were collected every 0.5 min (80 fractions), dried in a vacuum centrifuge, and dissolved in water to assess their activity against *M. xanthus* DK1622 and to identify whether some of the extracts could induce a clustering effect.

In the case of the *S. peucetius* supernatant, the initial procedure was similar. However, the activity was only present in the non-retained aqueous SPE flow-through, which was lyophilized and dissolved in the minimum amount of water for size exclusion chromatography on a Sephadex LH-20 column (70 g, 32 × 150 mm), which was eluted with 100% water at 1 mL/min to give 11 fractions of 20 mL each. Fractions retaining activity were concentrated to 3.5 mL and further fractionated by preparative HPLC on a Waters (Waters Corporation, Milford, MA, USA) T3 Atlantis OBD column (19.0 × 250 mm, 5 μm; 14 mL/min; UV detection at 210 and 280 nm) using an isocratic run with 100% water for 50 min, followed by a linear gradient of acetonitrile in water from 0 to 20% for 55 min and a wash step with 100% acetonitrile for 10 min. Fractions of 20 mL were collected (84 fractions) and 1/10 of the volume was dried in a vacuum centrifuge. All fractions obtained from the HPLC purification step were added to cultures of *M. xanthus* grown in 24-well plates containing cultures of 1 mL of liquid CYE. The cultures were grown at 28 °C for 3–5 days and the *M. xanthus* culture extracted with ethyl acetate containing 1% of formic acid was assayed against *S. cerevisiae*.

All fractions obtained throughout the fractionation of the *S. griseus* supernatant were assayed on solid CYE medium to check for *M. xanthus* mound formation.

### 4.6. Chromatographic Analysis

Analyses of the molecules produced in the different interactions on solid media were carried out by cutting the pieces of agar containing both spots of the microorganisms and extracting them with ethyl acetate containing 1% of formic acid. A piece of agar containing two spots of *M. xanthus* or two spots of the corresponding *Streptomyces* species was processed in the same way and used as a control.

For analyzing the liquid culture samples, 1 mL of the total liquid co-cultures, *M. xanthus/S. peucetius*, or 1 mL of the corresponding controls were extracted with 700 mL of ethyl acetate containing 1% of formic acid. The solvent was evaporated, and the residue was dissolved in 100 mL dimethyl sulfoxide: methanol (50:50). These samples were processed by UPLC or HPLC-MS as previously [14,40].

## 5. Conclusions

Competition for iron uptake plays a key role in the interaction of microorganisms and may change the developmental program as it is demonstrated in the interaction *S. griseus/M. xanthus* studied in this work. Future work studying the changes in *M. xanthus* gene expression under the aggregation induced by the siderophores demethylenenocardamine and nocardamine may give light to the reason for the changes in the growth pattern in complex media of this predatory bacteria.

## Figures and Tables

**Figure 1 ijms-24-15659-f001:**
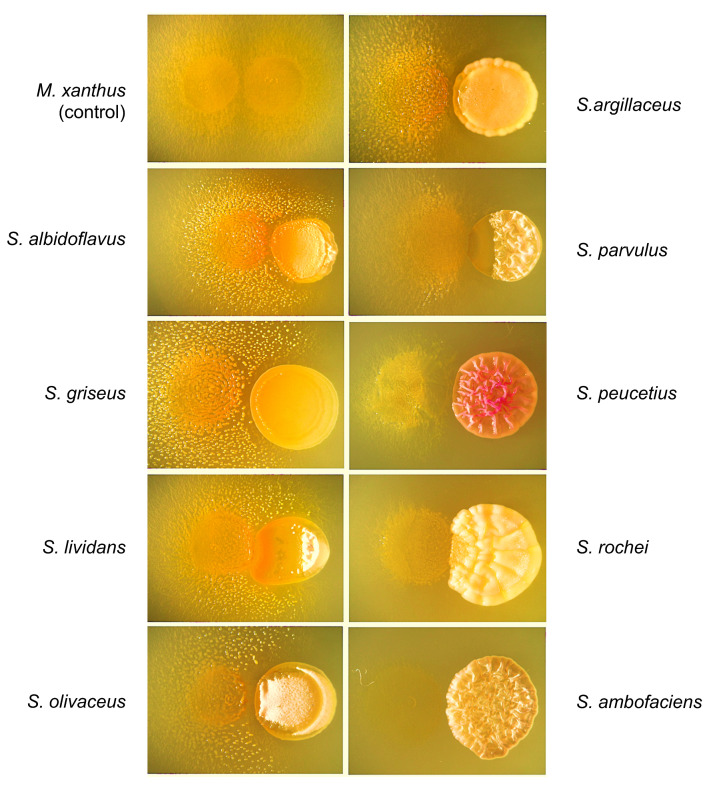
The predatory activity of *M. xanthus* DK1622 on different *Streptomyces* species on CYE plates. A spot of *M. xanthus* is on the left of each photograph; the spot on the right corresponds to the different *Streptomyces* prey indicated. Two drops of *M. xanthus* were used as a control. Pictures were taken from the top of the Petri dishes 7 days after the plates were inoculated. magnification was 6×.

**Figure 2 ijms-24-15659-f002:**
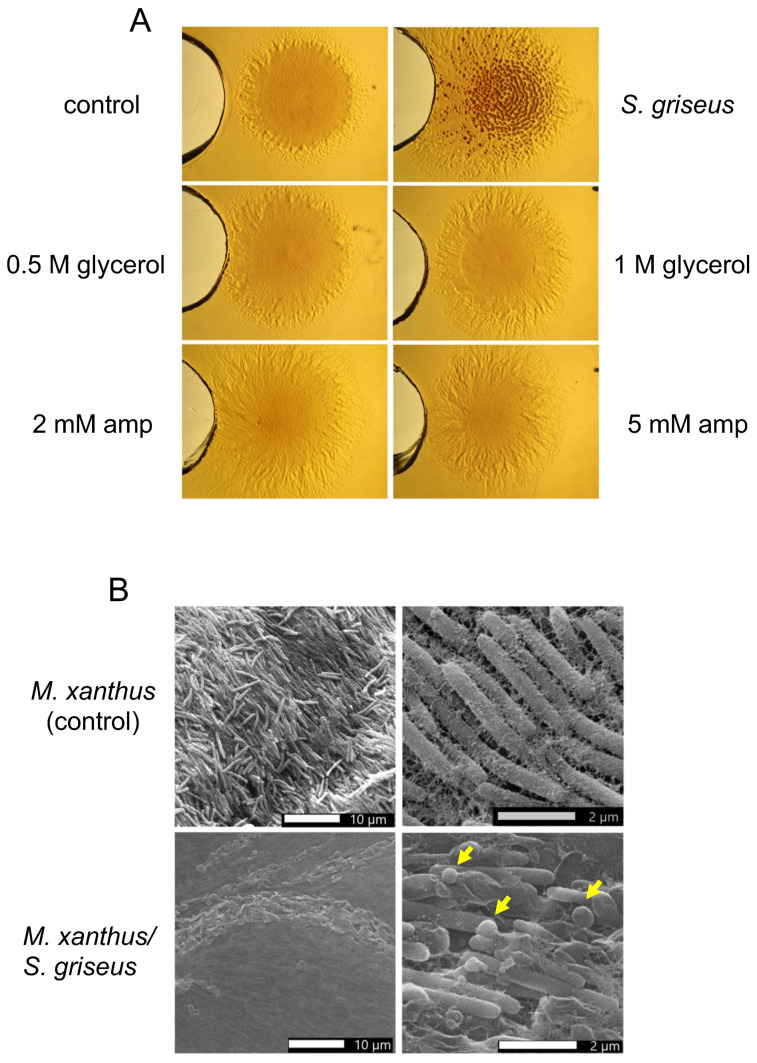
Induced aggregates of *M. xanthus* DK1622 on CYE plates. (**A**) Comparison of the effect of *S. griseus* supernatant grown in R5A-sucrose medium and different amounts of glycerol (0.5 M and 1 M) or ampicillin (amp: 2 and 5 mM) on the aggregation of *M. xanthus* DK1622 (the spot on the right side of each photograph); control (R5A-sucrose medium). magnification was 6×. The different compounds tested were added to the well on the left of each spot. (**B**) Scanning microscopy images of an axenic culture of *M. xanthus* (control) or of this strain grown in the presence of *S. griseus* supernatant *(M. xanthus/S. griseus)*. The yellow arrows indicate some of the myxospores observed.

**Figure 3 ijms-24-15659-f003:**
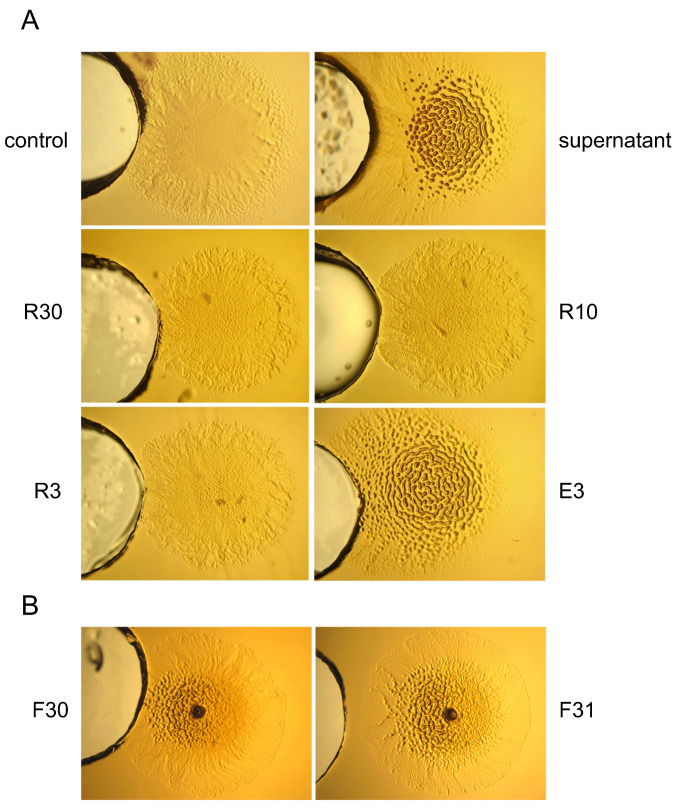
Identification of the active fractions of *S. griseus* supernatant that can induce aggregation in *M. xanthus DK1622* on CYE. (**A**) The effect of different fractions of the supernatant on *M. xanthus* behavior (drop on the right of each photograph). The different fractions tested were added to the well on the left of each spot of *M. xanthus*: control (R5A-sucrose), supernatant (supernatant of *S. griseus)*; R30 (retained in 30 kDa centricon); R10 (retained in 10 kDa centricon); R3 (retained in 3 kDa centricon); E3 (eluted in 3 kDa centricon). (**B**) The effect of the two active fractions obtained after HPLC purification of E3: F30 (Fraction 30) and F31 (Fraction 31). magnification was 6×.

**Figure 4 ijms-24-15659-f004:**
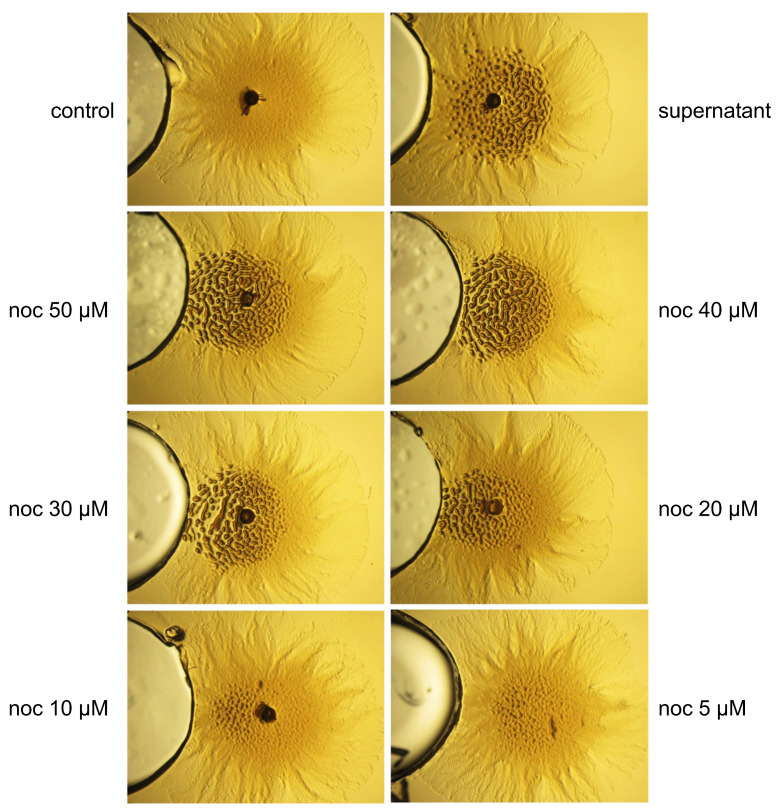
Effect of the siderophore nocardamine on *M. xanthus DK1622* on CYE. The effect of different amounts of nocardamine (noc 5, 10, 20, 30, 40, and 50 µM) on *M. xanthus* behavior (spot on the right of each photograph). The nocardamine was added to the well on the left of each spot of *M. xanthus*. The supernatant of *S. griseus* grown in R5A-sucrose medium was used as a positive control and R5A-sucrose as a negative control. magnification was 6×.

**Figure 5 ijms-24-15659-f005:**
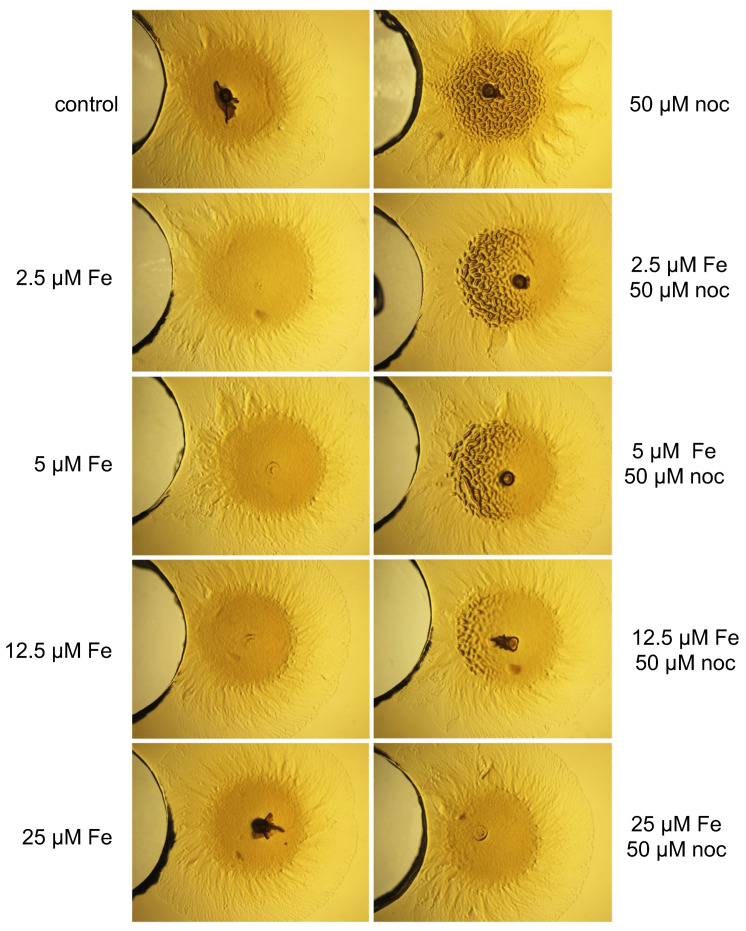
Effect of different amounts of Fe with or without the siderophore nocardamine on *M. xanthus DK1622* on CYE. The effect of different amounts of FeSO_4_ (Fe 2.5, 5, 12.5, and 25 μM) without nocardamine (noc; **left column**) and with (**right column**) 50 μM nocardamine (noc) on *M. xanthus* behavior (spot on the right of each photograph). Noc was added to the well on the left of each spot of *M. xanthus*. Nocardamine 50 μM was used as a positive control of group formation and H_2_O as a negative control. magnification was 6×.

**Figure 6 ijms-24-15659-f006:**
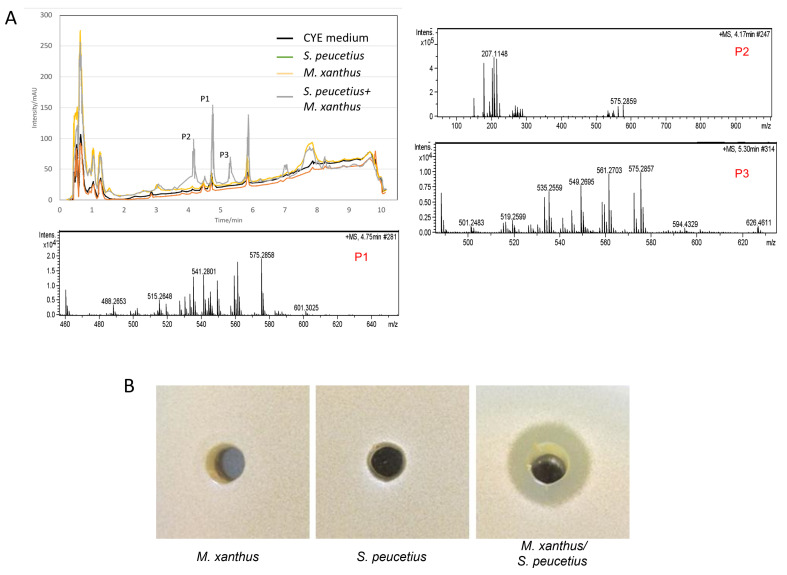
Analysis of the compounds produced by the interaction of *M. xanthus* DK1622 and *S. peucetius*. (**A**) HPLC-MS analysis of the compounds produced in the axenic cultures of *M. xanthus* and *S. peucetius* and the co-culture of *M. xanthus–S. peucetius* in liquid CYE UV-Vis absorbance (200–900 nm) and identification of P1, P2, and P3 by mass spectrometry. (**B**) Antifungal effect of the extracts of axenic cultures of *M. xanthus* and *S. peucetius*, and the co-culture of *M. xanthus/S. peucetius*.

**Figure 7 ijms-24-15659-f007:**
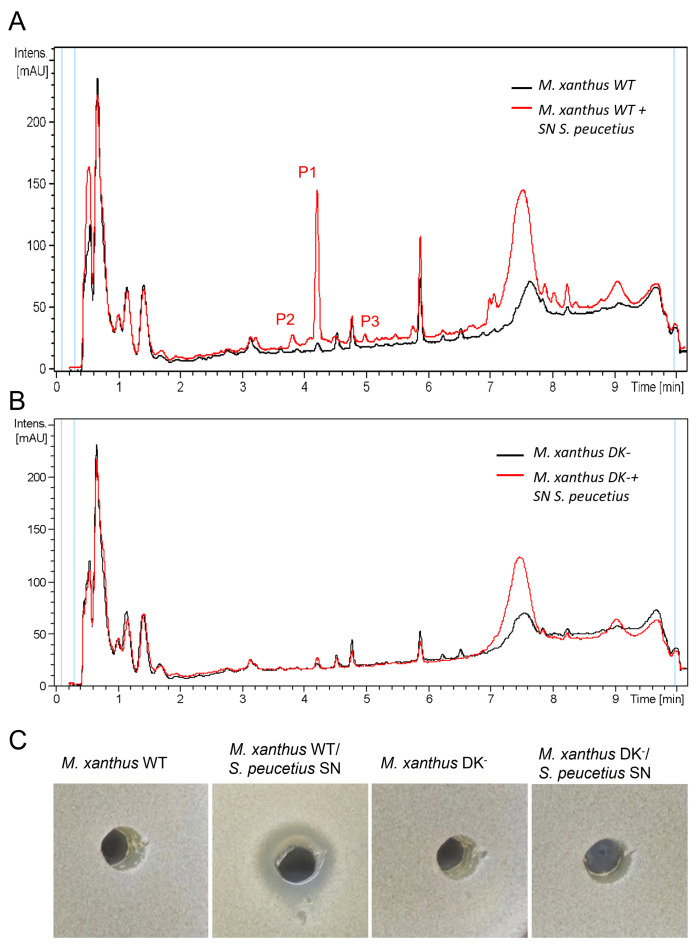
Analysis of the compounds produced by *M. xanthus* DK1050 and its DK-xanthenes minus strain PMΔRF_N1050 (DK^−^) derivative strain in the presence of *S. peucetius* supernatant (SN). HPLC analysis of the compounds produced under these culture conditions in the DK1050 (**A**) and in the (DK^−^) strain (**B**) by UV-Vis absorbance (200–900 nm). Antifungal effect of the extracts of *M. xanthus* DK1050 (wt) and its DK^-^ strain grown on the absence or presence of *S. peucetius* SN (**C**).

**Figure 8 ijms-24-15659-f008:**
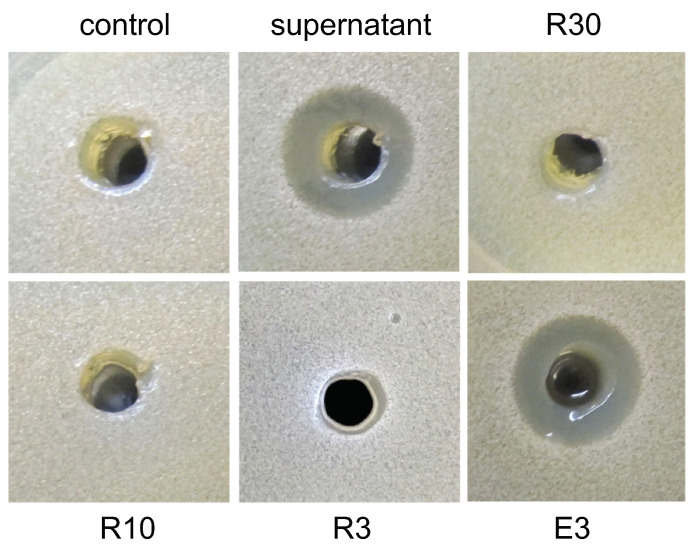
Antifungal activity of *M. xanthus* DK1622 cultures grown in CYE with different fractions of *S. peucetius* supernatant. R30 (retained in 30 kDa centricon); R10 (retained in 10 kDa centricon); R3 (retained in 3 kDa centricon); E3 (eluted in 3 kDa centricon); control: liquid R5A-sucrose medium.

## Data Availability

Not applicable.

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
