# Peer review of "Interactions of Different Streptomyces Species and Myxococcus xanthus Affect Myxococcus Development and Induce the Production of DK-Xanthenes"

_ijms, 2023, doi:10.3390/ijms242115659_

Round 1

Reviewer 1 Report

Comments and Suggestions for Authors

The article by Santamaria et al. describes the interactions between Myxococcus xanthus and various Streptomyces species by means of co-cultures in the laboratory. In support of these culture approaches and observation of the M. xanthus phenotypes, biochemical approaches enabled them to demonstrate i) the effect of siderophore production by S. griseus on the behaviour of M. xanthus and ii) the production of biomolecules (xanthenes) by M. xanthus in the presence of S. peucetius.

The article as a whole is well written, clear and follows a logical approach. The experiments are well performed and rigorously interpreted. I therefore support the publication of this article as it stands.

Minor comments

There is a discrepancy between the citation numeration in the introduction and in the references.

L. 92 Nicault et al. instead of Nicoult.

L.260-273: Is there any way of knowing whether the peaks observed in induction in solid and liquid media can be the same? Perhaps this should be specified in the text?

L324-329: Some information relates more to material and method section than to results, and perhaps does not need to be specified here.

L400-401: The sentence "In our study...development" seems badly formulated & "species" instead of "specie".

Author Response

Minor comments

Question: There is a discrepancy between the citation numeration in the introduction and in the references.   

Answer: This has been corrected.

The discrepancy was generated by the DOI number of the reference

Traxler, M.F.; Watrous, J.D.; Alexandrov, T.; Dorrestein, P.C.; Kolter, R. Interspecies interactions stimulate diversification of the Streptomyces coelicolor secreted metabolome. MBio 2013, 4, e00459-00413. doi:10.1128/mBio.00459-13.

That number appeared in a separate lane and originates the change of number of all the references after it. Thanks for detect this mistake generated by End-Note.

Question: L. 92 Nicault et al. instead of Nicoult.

Answer: This has been corrected

Question: L.260-273: Is there any way of knowing whether the peaks observed in induction in solid and liquid media can be the same? Perhaps this should be specified in the text?

Answer: Yes, the peaks induced on solid and in liquid are the same. This information was already described in the discussion part with the following sentences: “Furthermore, this S. peucetius / M. xanthus co-culture resulted in a dramatic increase in the production of DK-xanthenes by M. xanthus. The same induction was observed when S. peucetius supernatant, obtained from liquid R5A-sucrose cultures, was added to liquid CYE or CTT cultures of M. xanthus

Question: L324-329: Some information relates more to material and method section than to results, and perhaps does not need to be specified here.

Answer: These lanes have been eliminated due that the information was redundant with the information that appears in Materials and Methods.

Question: L400-401: The sentence "In our study...development" seems badly formulated & "species" instead of "specie".

Answer: This sentence has been changed and now reads

“In our study the siderophores produced by S. griseus control M. xanthus development”.

Reviewer 2 Report

Comments and Suggestions for Authors

The manuscript of Santamaria et al. entitled “Interactions of Different Streptomyces Species and Myxococcus xanthus Affect Myxococcus Development and Induce the Production of DK-xanthenes” adresses the question of the communication and reciprocal interactions between two specific microorganisms, Streptomyces (Gram +) and Myxococcus (Gram -).
The authors demonstrated that Mixococcus xanthus aggregates when siderophores are being produced by co-cultured Streptomyces species. However, in the meantime Myxococcus is growing more actively around and even over the siderophores producing Streptomyces species.
How do the authors explain this apparent contradiction? Is Myxococcus starved in iron and is this iron starvation triggers aggregation as well as the production of metabolites and/or enzymes able to lyse Streptomyces species? or is Myxococcus able to capture the iron loaded siderophores produced by the Streptomyces and is in fact not at all limited in iron?
The authors also demonstrated that a yet non-identified molecule produced by Streptomyces peucetius triggers the production of xanthene that has anti-fungal activity. It would be nice to determine whether xanthene has also an impact of S. peucetius growth and fitness.
The paper is interesting, rather well written and pleasant to read but that is the shame that the authors end their nice paper by insisting on a negative result. A more positive conclusion should be proposed.

However it would be nice if the authors could address the following pending questions:

-          On which criteria the 9 Streptomyces species were they chosen?

-          The authors mention that specialized metabolites biosynthetic pathways represent 8.5% of the genome of Myxococcus xanthus. It would be nice also to quote also what % of the Streptomyces genome they represent and what is the average size and the GC richness of the genome of the two bacteria.

-          Are the biosynthetic pathways of demethylenenocardamine and nocardamine identified in the S. griseus genome?

-          Are the biosynthetic pathways of xanthenes identified in the Myxococcus genome?

-          The authors should perhaps comment on why a depletion of iron would stimulate Actinorhoddin (ACT) biosynthesis. It is well known that iron is especially necessary for the functioning of the respiratory chain whose numerous enzymes bear iron-sulfur clusters. Interestingly a study indicated that ACT might have an anti-respiratory function (https://pubmed.ncbi.nlm.nih.gov/28298624/) besides its anti-oxidant function (https://www.ncbi.nlm.nih.gov/pmc/articles/PMC7244524/). These functions are thought to be linked to the ability of the ACT quinone groups to capture electrons. So in condition of iron depletion respiration should be tuned down and ACT (and other specialized metabolites with similar structure) might fulfil this role (https://pubmed.ncbi.nlm.nih.gov/32069930/).

-          Xanthene as ACT is constituted by aromatic rings and might have similar functions. So it would be nice to know whether in condition of co-culture S. peucetius is also producing a siderophore?

-          C is missing in the legend of Figure 7

Comments on the Quality of English Language

The quality of the english is good but some minor editorial corrections were proposed.

Author Response

The manuscript of Santamaria et al. entitled “Interactions of Different Streptomyces Species and Myxococcus xanthus Affect Myxococcus Development and Induce the Production of DK-xanthenes” adresses the question of the communication and reciprocal interactions between two specific microorganisms, Streptomyces (Gram +) and Myxococcus (Gram -).
Question: The authors demonstrated that Mixococcus xanthus aggregates when siderophores are being produced by co-cultured Streptomyces species. However, in the meantime Myxococcus is growing more actively around and even over the siderophores producing Streptomyces species.
How do the authors explain this apparent contradiction? Is Myxococcus starved in iron and is this iron starvation triggers aggregation as well as the production of metabolites and/or enzymes able to lyse Streptomyces species? or is Myxococcus able to capture the iron loaded siderophores produced by the Streptomyces and is in fact not at all limited in iron?.

Answer: Although we have not measured the mass of Myxococcus on the control and on the different interactions, the size (diameter) of the Myxococcus control colonies and the diameter of Myxococcus colony in all the cocultures that induce the aggregations are similar (Figure 1). However, it is clear that the diameter of the Myxococcus colonies in other cocultures as S. peucetius, S. rochei or S. ambofaciens is much smaller indicating the negative effect of these species over Myxococcus growth.  

The last possibility described by the reviewer is possible. In fact, Yamanaka et al (Microbiology 2005 151:2899-2905). )described that the production of Desferrioxamine E (nocardamine) produced by S. griseus stimulates the growth and differentiation of other Streptomyces (S. tanashiensis). However, it is clear that the addition of excess of iron to the plates (in presence of the siderophores) originates a normal growth of Myxococcus.

Question: The authors also demonstrated that a yet non-identified molecule produced by Streptomyces peucetius triggers the production of xanthene that has anti-fungal activity. It would be nice to determine whether xanthene has also an impact of S. peucetius growth and fitness.

Answer: We have never detected new compounds produced by S. peucetius when cocultured with M. xanthus, but we have never considered the possibility to use pure xanthenes over S. peucetius axenic cultures. Maybe we can try this assay for the future. Thanks for the suggestion.

Question: The paper is interesting, rather well written and pleasant to read but that is the shame that the authors end their nice paper by insisting on a negative result. A more positive conclusion should be proposed.

Answer: We must admit that we have insisted “too much” in the negative result of the no identification of the molecule(s) produced by S. peucetius. But we have done a lot of experiments trying to identify it. Some sentences considering this analysis have been eliminated from the manuscript.

Question: However it would be nice if the authors could address the following pending questions:

  • On which criteria the 9 Streptomyces species were they chosen?

Answer: The only criteria to select these species was that there were the only ones from public collections that we have in our laboratory collection.

-Question: The authors mention that specialized metabolites biosynthetic pathways represent 8.5% of the genome of Myxococcus xanthus. It would be nice also to quote also what % of the Streptomyces genome they represent and what is the average size and the GC richness of the genome of the two bacteria.

Answer: All the Streptomyces genomes have a GC content higher to 70%. And the part of their genomes that encode for specialized metabolites is higher to 12%. The genome of M.xanthus has a GG content of 69%.

-Question:  Are the biosynthetic pathways of demethylenenocardamine and nocardamine identified in the S. griseus genome?.

Answer: The sequence of the genome of the S. griseus strain used in this work is not in the NCBI genome data bank. However the strain S. griseus NBCR13350 is sequenced (https://www.ncbi.nlm.nih.gov/datasets/taxonomy/455632/). Antismash analysis of this genome identified the cluster for the siderophore desferrioxamine B (https://mibig.secondarymetabolites.org/repository/BGC0000941/index.html#r1c1) that borrow a high similarity with the Des cluster studied and cloned previously from Streptomyces coelicolor. Previous work in S. coelicolor identified that the same biosynthetic cluster encodes desferrioxamine E (Nocardamine) and desferrioxamine B (https://pubmed.ncbi.nlm.nih.gov/15600304/ and https://pubmed.ncbi.nlm.nih.gov/17074905/).

As far as we know, no biosynthetic cluster has been identified for demethylenenocardamine in Streptomyces. In fact, the references to this compound are limited and have been described in a very limited number of Streptomyces from different habitats. One of these Streptomyces producers of demethylenenocardamine was isolated by our group from feces from the larvae of the Longhorn beetle Cerambix welensii (https://pubmed.ncbi.nlm.nih.gov/33339339/).

-Question: Are the biosynthetic pathways of xanthenes identified in the Myxococcus genome?

Answer: Yes, the biosynthetic pathway of DK-xanthenes from M. xanthus was published in 2006

https://www.pnas.org/doi/10.1073/pnas.0606039103

-Question: The authors should perhaps comment on why a depletion of iron would stimulate Actinorhoddin (ACT) biosynthesis. It is well known that iron is especially necessary for the functioning of the respiratory chain whose numerous enzymes bear iron-sulfur clusters. Interestingly a study indicated that ACT might have an anti-respiratory function (https://pubmed.ncbi.nlm.nih.gov/28298624/) besides its anti-oxidant function (https://www.ncbi.nlm.nih.gov/pmc/articles/PMC7244524/). These functions are thought to be linked to the ability of the ACT quinone groups to capture electrons. So in condition of iron depletion respiration should be tuned down and ACT (and other specialized metabolites with similar structure) might fulfil this role (https://pubmed.ncbi.nlm.nih.gov/32069930/).

Answer: It is clear that the iron plays a key role in antibiotic production and the molecular mechanism responsible to this effect is an active field of research. The effect of iron limitation in actinorhodin production is well described in the paper of Lee and collaborators (https://pubmed.ncbi.nlm.nih.gov/31992858/) and in some way complemented by the studies of Dr. Virolle’s group. So, in the interaction Myxococcus / S. coelicolor iron, that is needed for respiratory proteins, is sequestered and this may generate more ROS in S. coelicolor that induce Actinorhodin to counteract signals from M. xanthus.

-Question: Xanthene as ACT is constituted by aromatic rings and might have similar functions. So it would be nice to know whether in condition of co-culture S. peucetius is also producing a siderophore?

Answer:As described in the manuscript, the coculture M. xanthus /S. peucetius induce clearly the production of DK-xanthenes by M. xanthus but we cannot discard that other minority compound is produced and we have not detected it in our study.

-Question:  C is missing in the legend of Figure 7

Answer:This has been corrected

Answer: All the corrections of English have been incorporated in the text. Thanks for all the suggestions.

Question: Comments on the Quality of English Language

Minor editorial changes recommended in bold:

The quality of the english is good but some minor editorial corrections were proposed.

Answer: All the corrections of English have been incorporated in the text. Thanks for all the suggestions.

Reviewer 3 Report

Comments and Suggestions for Authors

Please find attached the manuscript with comments and suggestions.

Author Response

Answer: Please see enclosed PDF of the manuscript on which we have answered point by point to all the questions. All the corrections have been incorporated in the corrected word version  enclosed.

All the questions have been answered in the PDF file attached and all the corrections have been incorporated in the manuscript.

About the references in the results part we consider that the references included are necessary for the reader to understand the reason why the different experiments are done. And more so in this Journal in which the materials and methods go at the end of the article.

As an example I selected IJMS volume 24 Issue1 and I open the  first two articles, Both have references in the results part:

https://www.mdpi.com/1422-0067/24/1/894

Round 2

Reviewer 3 Report

Comments and Suggestions for Authors

Dear authors, I accept the manuscript in the current form, Thank you for improving the manuscript according to the suggestions and for replying to all my questions.